# Learning Adversarial Linear Mixture Markov Decision Processes with Bandit Feedback and Unknown Transition

**Canzhe Zhao[1], Ruofeng Yang[1], Baoxiang Wang[2], Shuai Li[1]***

[1]John Hopcroft Center for Computer Science, Shanghai Jiao Tong University
[2]School of Data Science, The Chinese University of Hong Kong, Shenzhen
{canzhezhao,wanshuiyin,shuaili8}@sjtu.edu.cn
bxiangwang@cuhk.edu.cn

## Abstract

We study reinforcement learning (RL) with linear function approximation, unknown transition, and adversarial losses in the bandit feedback setting. Specifically, the unknown transition probability function is a linear mixture model (Ayoub et al., 2020; Zhou et al., 2021; He et al., 2022) with a given feature mapping, and the learner only observes the losses of the experienced state-action pairs instead of the whole loss function. We propose an efficient algorithm LSUOB-REPS which achieves $\widetilde{O}(dS^2\sqrt{K} + \sqrt{HSAK})$ regret guarantee with high probability, where $d$ is the ambient dimension of the feature mapping, $S$ is the size of the state space, $A$ is the size of the action space, $H$ is the episode length and $K$ is the number of episodes. Furthermore, we also prove a lower bound of order $\Omega(dH\sqrt{K} + \sqrt{HSAK})$ for this setting. To the best of our knowledge, we make the first step to establish a provably efficient algorithm with a sublinear regret guarantee in this challenging setting and solve the open problem of He et al. (2022).

## 1 Introduction

Reinforcement learning (RL) has achieved significant empirical success in the fields of games, control, robotics and so on. One of the most notable RL models is the Markov decision process (MDP) (Feinberg, 1996). For tabular MDP with finite state and action spaces, the nearly minimax optimal sample complexity is achieved in discounted MDPs with a generative model (Azar et al., 2013). Without the access of a generative model, the nearly minimax optimal sample complexity is established in tabular MDPs with finite horizon (Azar et al., 2017) and in tabular MDPs with infinite horizon (He et al., 2021b; Tossou et al., 2019). However, in real applications of RL, the state and action spaces are possibly very large and even infinite. In this case, the tabular MDPs are known to suffer the curse of dimensionality. To overcome this issue, recent works consider studying MDPs under the assumption of function approximation to reparameterize the values of state-action pairs by embedding the state-action pairs in some low-dimensional space via given feature mapping. In particular, linear function approximation has gained extensive research attention. Amongst these works, linear mixture MDPs (Ayoub et al., 2020) and linear MDPs (Jin et al., 2020b) are two of the most popular MDP models with linear function approximation. Recent works have attained the minimax optimal regret guarantee $\widetilde{O}(dH\sqrt{KH})$ in both linear mixture MDPs (Zhou et al., 2021) and linear MDPs (Hu et al., 2022) with stochastic losses.

Though significant advances have emerged in learning tabular MDPs and MDPs with linear function approximation under stochastic loss functions, in real applications of RL, the loss functions may not be fixed or sampled from some certain underlying distribution. To cope with this challenge, Even-Dar et al. (2009); Yu et al. (2009) make the first step to study learning adversarial MDPs, where the loss functions are chosen adversarially and may change arbitrarily between each step. Most works in this line of research focus on learning adversarial tabular MDPs (Neu et al., 2010a;b; 2012; Arora

---

*Corresponding author.

Table 1: Comparisons of regret bounds with most related works studying adversarial tabular and linear mixture MDPs with unknown transitions. $K$ is the number of episodes, $d$ is the ambient dimension of the feature mapping, $S$ is the size of the state space, $A$ is the size of the action space, and $H$ is the episode length.

| Algorithm | Model | Feedback | Regret |
|---|---|---|---|
| Shifted Bandit UC-O-REPS (Rosenberg & Mansour, 2019a) | Tabular MDPs | Bandit Feedback | $\widetilde{O}\left(H^{3/2}SA^{1/4}K^{3/4}\right)$ |
| UOB-REPS (Jin et al., 2020a) | Tabular MDPs | Bandit Feedback | $\widetilde{O}\left(HS\sqrt{AK}\right)$ |
| OPPO (Cai et al., 2020) | Linear Mixture MDPs | Full-information | $\widetilde{O}\left(dH^2\sqrt{K}\right)$ |
| POWERS (He et al., 2022) | Linear Mixture MDPs | Full-information | $\widetilde{O}\left(dH^{3/2}\sqrt{K}\right)$ |
| LSUOB-REPS **(Ours)** | Linear Mixture MDPs | Bandit Feedback | $\widetilde{O}\left(dS^2\sqrt{K}+\sqrt{HSAK}\right)$ $\Omega\left(dH\sqrt{K}+\sqrt{HSAK}\right)$ |

et al., 2012; Zimin & Neu, 2013; Dekel & Hazan, 2013; Dick et al., 2014; Rosenberg & Mansour, 2019a;b; Jin & Luo, 2020; Jin et al., 2020a; Shani et al., 2020; Chen et al., 2021; Ghasemi et al., 2021; Rosenberg & Mansour, 2021; Jin et al., 2021b; Dai et al., 2022; Chen et al., 2022a). In contrast, most recent advances regarding learning adversarial MDPs with linear function approximation require some stringent assumptions and we are still far from understanding it well. Specifically, Cai et al. (2020); He et al. (2022) study learning episodic adversarial linear mixture MDPs with unknown transition but under *full-information* feedback and Neu & Olkhovskaya (2021) study learning episodic adversarial linear MDPs under bandit feedback but with *known* transition. In the more challenging setting with both unknown transition and bandit feedback, Luo et al. (2021b) make the first step to establish a sublinear regret guarantee $\widetilde{O}(K^{6/7})$ in adversarial linear MDPs under the assumption that there exists an exploratory policy and Luo et al. (2021a) (an improved version of Luo et al. (2021b)) obtain a regret guarantee $\widetilde{O}(K^{14/15})$ in the same setting but without access to an exploratory policy. Therefore, a natural question remains open:

*Does there exist a provably efficient algorithm with $\widetilde{O}(\sqrt{K})$ regret guarantee for RL with linear function approximation under unknown transition, adversarial losses and bandit feedback?*

In this paper, we give an affirmative answer to this question in the setting of linear mixture MDPs and hence solve the open problem of He et al. (2022). Specifically, we propose an algorithm termed LSUOB-REPS for adversarial linear mixture MDPs with unknown transition and bandit feedback. To remove the need for the full-information feedback of the loss function required by policy-optimization-based methods (Cai et al., 2020; He et al., 2022), LSUOB-REPS extends the general ideas of occupancy-measure-based methods for adversarial tabular MDPs with unknown transition (Jin et al., 2020a; Rosenberg & Mansour, 2019a;b; Jin et al., 2021b). Specifically, inspired by the UC-O-REPS algorithm (Rosenberg & Mansour, 2019b;a), LSUOB-REPS maintains a confidence set of the unknown transition and runs online mirror descent (OMD) over the space of occupancy measures induced by all the statistically plausible transitions within the confidence set to handle the unknown transition. The key difference is that we need to build some sort of least-squares estimate of the transition parameter and its corresponding confidence set to leverage the transition structure of the linear mixture MDPs. Previous works studying linear mixture MDPs (Ayoub et al., 2020; Cai et al., 2020; He et al., 2021a; Zhou et al., 2021; He et al., 2022; Wu et al., 2022; Chen et al., 2022b; Min et al., 2022) use the state values as the regression targets to learn the transition parameter. This method is critical to construct the optimistic estimate of the state-action values and attain the final regret guarantee. In this way, however, it is difficult to control the estimation error between the occupancy measure computed by OMD and the one that the learner really takes.

To cope with this issue, we use the transition information of the next-states as the regression targets to learn the transition parameter. In particular, we pick a certain next-state, which we call the *imaginary*

next-state, and use its transition information as the regression target (see Section 4.1 for details). In this manner, we are able to control the occupancy measure error efficiently. Besides, since the true transition is unknown, the true occupancy measure taken by the learner is also unknown and it is infeasible to construct an unbiased loss estimator using the standard importance weighting method. To this end, we use the upper occupancy measure (Jin et al., 2020a) together with a hyperparameter to conduct implicit exploration (Neu, 2015) to construct an optimistically biased loss estimator. Finally, we prove the $\widetilde{O}(dS^2\sqrt{K} + \sqrt{HSAK})$ high probability regret guarantee of LSUOB-REPS, where $S$ is the size of the state space, $A$ is the size of the action space, $H$ is the episode length, $d$ is the dimension of the feature mapping, and $K$ is the number of the episodes. Further, we also prove a lower bound of order $\Omega(dH\sqrt{K} + \sqrt{HSAK})$, which matches the upper bound in $d$, $K$ and $A$ up to logarithmic factors (please see Table 1 for the comparisons between our results and previous ones). Though the upper bound does not match lower bounds in $S$, we establish the first provably efficient algorithm with $\widetilde{O}(\sqrt{K})$ regret guarantee for learning adversarial linear mixture MDPs under unknown transition and bandit feedback.

## 2 RELATED WORK

**RL with Linear Function Approximation**    To permit efficient learning in RL with large state-action space, recent works have focused on RL algorithms with linear function approximation. In general, these works can be categorized into three lines. The first line uses the low Bellman-rank assumption (Jiang et al., 2017; Dann et al., 2018; Sun et al., 2019; Du et al., 2019; Jin et al., 2021a), which assumes the Bellman error matrix has a low-rank factorization. Besides, Du et al. (2021) consider a similar but more general assumption called bounded bilinear rank. The second line considers the linear MDP assumption (Yang & Wang, 2019; Jin et al., 2020b; Du et al., 2020; Zanette et al., 2020a; Wang et al., 2020; 2021; He et al., 2021a; Hu et al., 2022), where both the transition probability function and the loss function can be parameterized as linear functions of given state-action feature mappings. In particular, Jin et al. (2020b) propose the first statistically and computationally efficient algorithm with $\widetilde{O}(H^2\sqrt{d^3K})$ regret guarantee. Hu et al. (2022) further improve this result by using a weighted ridge regression and a Bernstein-type exploration bonus and obtain the minimax optimal regret bound $\widetilde{O}(dH\sqrt{KH})$. Zanette et al. (2020b) consider a weaker assumption called low inherent Bellman error, where the Bellman backup is linear in the underlying parameter up to some misspecification errors. The last line of works considers the linear mixture MDP assumption (Ayoub et al., 2020; Zhang et al., 2021; Zhou et al., 2021; He et al., 2021a; Zhou & Gu, 2022; Wu et al., 2022; Min et al., 2022), in which the transition probability function is linear in some underlying parameter and a given feature mapping over state-action-next-state triples. Amongst these works, Zhou et al. (2021) obtain the minimax optimal regret bound $\widetilde{O}(dH\sqrt{KH})$ in the inhomogeneous episodic linear mixture MDP setting. In this work, we also focus on linear mixture MDPs.

**RL with Adversarial Losses**    Learning tabular RL with adversarial losses has been well-studied (Neu et al., 2010a;b; 2012; Arora et al., 2012; Zimin & Neu, 2013; Dekel & Hazan, 2013; Dick et al., 2014; Rosenberg & Mansour, 2019a;b; Jin & Luo, 2020; Jin et al., 2020a; Shani et al., 2020; Chen et al., 2021; Ghasemi et al., 2021; Rosenberg & Mansour, 2021; Jin et al., 2021b; Dai et al., 2022; Chen et al., 2022a). Generally, these results fall into two categories. The first category studies adversarial RL using occupancy-measure-based methods. In particular, with known transition, Zimin & Neu (2013) propose the O-REPS algorithm, which achieves (near) optimal regret $\widetilde{O}(H\sqrt{K})$ with full-information feedback and $\widetilde{O}(\sqrt{HSAK})$ with bandit feedback respectively. With unknown transition and full-information feedback, Rosenberg & Mansour (2019b) propose UC-O-REPS algorithm, and achieve $\widetilde{O}(HS\sqrt{AK})$ regret guarantee. When the transition is unknown, and only the bandit feedback is available, Rosenberg & Mansour (2019a) propose the bounded bandit UC-O-REPS algorithm and achieve $\widetilde{O}(HS\sqrt{AK}/\alpha)$ regret bound with the assumption that all states are reachable with probability $\alpha$. Without this assumption, Rosenberg & Mansour (2019a) only achieve $\widetilde{O}(H^{3/2}SA^{1/4}K^{3/4})$ regret bound. Under the same setting but without the strong assumption of Rosenberg & Mansour (2019a), Jin et al. (2020a) develop the UOB-REPS algorithm, which uses a tight confidence set for transition function and a new biased loss estimator and achieves $\widetilde{O}(HS\sqrt{AK})$ regret bound. Besides, we remark that the existing tightest lower bound is $\Omega(H\sqrt{SAK})$ for the unknown transition and full-information feedback setting (Jin et al., 2018). The second category

for learning adversarial RL is the policy-optimization-based method (Neu et al., 2010a; Shani et al., 2020; Luo et al., 2021b; Chen et al., 2022a), which aims to directly optimize the policies. In this line of research, with known transition and bandit feedback, Neu et al. (2010b) propose OMDP-BF algorithm and achieve a regret of order $\widetilde{O}(K^{2/3})$. Recently, Shani et al. (2020) establish the POMD algorithm and attain a $\widetilde{O}(\sqrt{S^2AH^4}K^{2/3})$ regret bound for unknown transition and bandit feedback setting, which is further improved to $\widetilde{O}\left(H^2S\sqrt{AK} + H^4\right)$ by Luo et al. (2021b) in the same setting.

Recent advances have also emerged in learning adversarial RL with linear function approximation (Cai et al., 2020; He et al., 2022; Neu & Olkhovskaya, 2021; Luo et al., 2021a;b). Most of these works study this problem using policy-optimization-based methods (Cai et al., 2020; Luo et al., 2021a;b; He et al., 2022). Remarkably, He et al. (2022) achieve the (near) optimal $\widetilde{O}(dH^{3/2}\sqrt{K})$ regret bound in adversarial linear mixture MDPs in unknown transition but full-information feedback setting. With bandit feedback but known transition, Neu & Olkhovskaya (2021) obtain a $\widetilde{O}(\sqrt{dHK})$ regret guarantee in linear MDPs by using an occupancy-measure-based algorithm called Q-REPS. Luo et al. (2021a) make the first step to establish a sublinear regret guarantee $\widetilde{O}\left(d^2H^4K^{14/15}\right)$ in adversarial linear MDPs with unknown transition and bandit feedback.

## 3 PRELIMINARIES

In this section, we present the preliminaries of episodic linear mixture MDPs under adversarial losses.

**Inhomogeneous, episodic adversarial MDPs** An inhomogeneous, episodic adversarial MDP is denoted by a tuple $\mathcal{M} = (\mathcal{S}, \mathcal{A}, H, \{P_h\}_{h=0}^{H-1}, \{\ell_k\}_{k=1}^{K})$, where $\mathcal{S}$ is the finite state space with cardinality $|\mathcal{S}| = S$, $\mathcal{A}$ is the finite action space with cardinality $|\mathcal{A}| = A$, $H$ is the length of each episode, $P_h : \mathcal{S} \times \mathcal{A} \times \mathcal{S} \to [0,1]$ is the transition probability function with $P_h(s'|s,a)$ being the probability of transferring to state $s'$ from state $s$ and taking action $a$ at stage $h$, and $\ell_k : \mathcal{S} \times \mathcal{A} \to [0,1]$ is the loss function for episode $k$ chosen by the adversary. Without loss of generality, we assume that the MDP has a layered structure, satisfying the following conditions:

- The state space $\mathcal{S}$ is constituted by $H + 1$ disjoint layers $\mathcal{S}_0, \ldots, \mathcal{S}_H$ satisfying $\mathcal{S} = \bigcup_{h=0}^{H} \mathcal{S}_h$ and $\mathcal{S}_i \bigcap \mathcal{S}_j = \emptyset$ for $i \neq j$.
- $\mathcal{S}_0$ and $\mathcal{S}_H$ are singletons, *i.e.*, $\mathcal{S}_0 = \{s_0\}$ and $\mathcal{S}_H = \{s_H\}$.
- Transitions can only occur between consecutive layers. Formally, let $h(s)$ represent the index of the layer to which state $s$ belongs, then $\forall s' \notin \mathcal{S}_{h(s)+1}$ and $\forall a \in \mathcal{A}$, $P_{h(s)}(s'|s,a) = 0$.

These assumptions are standard in previous works (Zimin & Neu, 2013; Rosenberg & Mansour, 2019b;a; Jin et al., 2020a; Jin & Luo, 2020; Jin et al., 2021b; Neu & Olkhovskaya, 2021). They are not necessary for our analysis but can simplify the notations. However, we remark that our layer structure assumption is slightly more general than it in previous works, which assume homogeneous transition functions (*i.e.*, $P_0 = P_1 = \ldots = P_{H-1}$). Hence they require $\forall s' \notin \mathcal{S}_{h(s)+1}$ and $\forall a \in \mathcal{A}$, $P_h(s'|s,a) = 0$ for all $h = 0, \ldots, H - 1$. Besides, in our formulation, due to the layer structure, $P_h(\cdot|s,a)$ will actually never affect the transitions in the MDP if $h \neq h(s)$. Hence, with slightly abuse of notation, we define $P := \{P_h\}_{h=0}^{H-1}$ and write $P(\cdot|s,a) = P_{h(s)}(\cdot|s,a)$.

The interaction protocol between the learner and the environment is given as follows. Ahead of time, the environment decides an MDP, and the learner only knows the state space $\mathcal{S}$, the layer structure, and the action space $\mathcal{A}$. The interaction proceeds in $K$ episodes. At the beginning of episode $k$, the adversary chooses a loss function $\ell_k$ probably based on the history information before episode $k$. Meanwhile, the learner chooses a stochastic policy $\pi_k : \mathcal{S} \times \mathcal{A} \to [0,1]$ with $\pi_k(a|s)$ being the probability of taking $a$ at state $s$. Starting from the initial state $s_{k,0} = s_0$, the learner repeatedly selects action $a_{k,h}$ sampled from $\pi_k(\cdot|s_{k,h})$, suffers loss $\ell_k(s_{k,h}, a_{k,h})$ and transits to the next state $s_{k,h+1}$ which is drawn from $P(\cdot|s_{k,h}, a_{k,h})$ for $h = 0, ..., H - 1$, until reaching the terminating state $s_{k,H} = s_H$. At the end of episode $k$, the learner only observes bandit feedback, *i.e.*, the learner only observes the loss for each visited state-action pair: $\{\ell_k(s_{k,h}, a_{k,h})\}_{h=0}^{H-1}$. For any $(s,a) \in \mathcal{S} \times \mathcal{A}$, the state-action value $Q_{k,h}(s,a)$ and state value $V_{k,h}(s)$ are defined as follows: $Q_{k,h}(s,a) = \mathbb{E}\left[\sum_{j=h}^{H-1} \ell_k(s_{k,j}, a_{k,j}) \Big| \pi, P, (s_{k,h}, a_{k,h}) = (s,a)\right]$ and $V_{k,h}(s) = \mathbb{E}_{a \sim \pi(\cdot|s)}[Q_{k,h}(s,a)]$.

We denote the expected loss of an policy $\pi$ in episode $k$ by $\ell_k(\pi) = \mathbb{E}\left[\sum_{h=0}^{H-1} \ell_k(s_{k,h}, a_{k,h}) \mid P, \pi\right]$, where the trajectory $\{(s_{k,h}, a_{k,h})\}_{h=0}^{H-1}$ is generated by executing policy $\pi$ under transition function $P$. The goal of the learner is to minimize the regret compared with $\pi^*$, defined as

$$R(K) = \sum_{k=1}^{K} \ell_k(\pi_k) - \sum_{k=1}^{K} \ell_k(\pi^*),$$

where $\pi^* \in \operatorname{argmin}_{\pi \in \Pi} \sum_{k=1}^{K} \ell_k(\pi)$ is the optimal policy and $\Pi$ is the set of all stochastic policies.

**Linear mixture MDPs** We consider a special class of MDPs called *linear mixture MDPs* (Ayoub et al., 2020; Cai et al., 2020; Zhou et al., 2021; He et al., 2022) where the transition probability function is linear in a known feature mapping $\phi : \mathcal{S} \times \mathcal{A} \times \mathcal{S} \to \mathbb{R}^d$. The formal definition of linear mixture MDPs is given as follows.

**Definition 1.** $\mathcal{M} = (\mathcal{S}, \mathcal{A}, H, \{P_h\}_{h=0}^{H-1}, \{\ell_k\}_{k=1}^K)$ *is called an inhomogeneous, episodic $B$-bounded linear mixture MDP if* $\|\phi(s'|s, a)\|_2 \leq 1$ *and there exist vectors* $\boldsymbol{\theta}_h^* \in \mathbb{R}^d$ *such that* $P_h(s'|s, a) = \langle \phi(s'|s, a), \boldsymbol{\theta}_h^* \rangle$, *and* $\|\boldsymbol{\theta}_h^*\|_2 \leq B$, $\forall (s, a, s') \in \mathcal{S} \times \mathcal{A} \times \mathcal{S}$ *and* $h = 0, 1, \ldots, H-1$.

We note that the regularity assumption on the feature mapping $\phi(\cdot|\cdot, \cdot)$ in this work is slightly different from it of Zhou et al. (2021); He et al. (2022). In particular, they assume $\|\phi_G(s, a)\|_2 \leq 1$ for any $(s, a) \in \mathcal{S} \times \mathcal{A}$ and any bounded function $G : \mathcal{S} \to [0, 1]$, where $\phi_G(s, a) = \sum_{s'} \phi(s'|s, a)G(s')$. One can see that our assumption is slightly more general than theirs.

**Notation** For a vector $\boldsymbol{x}$ and a matrix $\boldsymbol{A}$, we use $\boldsymbol{x}(i)$ to denote the $i$-th coordinate of $\boldsymbol{x}$ and use $\boldsymbol{A}(i, :)$ to denote the $i$-th row of $\boldsymbol{A}$. Let $o_{i,j} = (s_{i,j}, a_{i,j}, \ell_i(s_{i,j}, a_{i,j}))$ be the observation of the learner at episode $i$ and stage $j$. We denote by $\mathcal{F}_{k,h}$ the $\sigma$-algebra generated by $\{o_{1,0}, \ldots, o_{1,H-1}, o_{2,0}, \ldots, o_{k,0}, \ldots, o_{k,h}\}$. For simplicity, we abbreviate $\mathbb{E}[\cdot|\mathcal{F}_{k,h}]$ as $\mathbb{E}_{k,h}[\cdot]$. The notation $\widetilde{O}(\cdot)$ in this work hides all the logarithmic factors.

### 3.1 OCCUPANCY MEASURES

To solve the MDPs with online learning techniques, we consider using the concept of *occupancy measures* (Altman, 1998). Specifically, for some policy $\pi$ and a transition probability function $P$, the occupancy measure $q^{P,\pi} : \mathcal{S} \times \mathcal{A} \to [0, 1]$ induced by $P$ and $\pi$ is defined as $q^{P,\pi}(s, a) = \Pr[(s_h, a_h) = (s, a)|P, \pi]$, where $h = h(s)$ is the index of the layer of state $s$. Hence $q^{P,\pi}(s, a)$ indicates the probability of visiting state-action pair under policy $\pi$ and transition $P$. In what follows, we drop the dependence of an occupancy measure on $P$ and $\pi$ when it is clear from the context.

Due to its definition, a valid occupancy measure $q$ satisfies the following two conditions. First, since one and only one state in each layer will be visited in an episode in a layered MDP, $\forall h = 0, \ldots, H-1, \sum_{(s,a) \in \mathcal{S}_h \times \mathcal{A}} q(s, a) = 1$. Second, $\forall h = 1, \ldots, H-1$, and $\forall s \in \mathcal{S}_h$, $\sum_{(s',a') \in \mathcal{S}_{h-1} \times \mathcal{A}} q(s', a') P(s|s', a') = \sum_{a \in \mathcal{A}} q(s, a)$. With slightly abuse of notation, we write $q(s) = \sum_{a \in \mathcal{A}} q(s, a)$. For a given occupancy measure $q$, one can obtain its induced policy by $\pi^q(a|s) = q(s, a)/q(s)$. Fixing a transition function $P$ of interest, we denote by $\Delta(P)$ the set of all the valid occupancy measures induced by $P$ and some policy $\pi$. Then the regret can be rewritten as

$$R(K) = \sum_{k=1}^{K} \langle q^{P,\pi_k} - q^*, \ell_k \rangle, \tag{1}$$

where $q^* = q^{P,\pi^*} \in \Delta(P)$ is the optimal occupancy measure induced by $\pi^*$.

## 4 ALGORITHM

In this section, we introduce the proposed LSUOB-REPS algorithm, detailed in Algorithm 1. In general, LSUOB-REPS maintains a ellipsoid confidence set of the unknown transition parameter (Section 4.1). Meanwhile, it constructs an optimistically biased loss estimator and runs OMD over the space of the occupancy measures induced by the ellipsoid confidence set to update the occupancy measure (Section 4.2).

---

**Algorithm 1** Least Squares Upper Occupancy Bound Relative Entropy Policy Search (LSUOB-REPS)

1: **Input:** state space $\mathcal{S}$, action space $\mathcal{A}$, episode number $K$, learning rate $\eta$, exploration parameter $\gamma$, regression regularization parameter $\lambda$, and confidence parameter $\delta$
2: **Initialization:** Initialize confidence set $\mathcal{P}_1$ as the set of all transition functions. For all $h = 0, ..., H-1$ and all $(s, a) \in \mathcal{S}_h \times \mathcal{A}$, initialize $\boldsymbol{M}_{0,h} = \lambda \boldsymbol{I}$, occupancy measure $\widehat{q}_1(s, a) = \frac{1}{S_k \times A}$ and policy $\pi_1 = \pi^{\widehat{q}_1}$.
3: **for** $k = 1, 2, \ldots, K$ **do**
4:     **for** $h = 0, 1, \ldots, H-1$ **do**
5:         Take action $a_{k,h} \sim \pi_k(\cdot|s_{k,h})$.
6:         Set the imaginary next state $s'_{k,h+1} \in \text{argmax}_{s \in \mathcal{S}_{h+1}} \|\boldsymbol{\phi}(s|s_{k,h}, a_{k,h})\|_{\boldsymbol{M}^{-1}_{k-1,h}}$.
7:         Observe true next state $s_{k,h+1} \sim P_h(\cdot|s_{s,h}, a_{k,h})$ and loss $\ell_k(s_{k,h}, a_{k,h})$.
8:         $\boldsymbol{M}_{k,h} = \boldsymbol{M}_{k-1,h} + \boldsymbol{\phi}(s'_{k,h+1}|s_{k,h}, a_{k,h})\boldsymbol{\phi}(s'_{k,h+1}|s_{k,h}, a_{k,h})^{\top}$.
9:         $\boldsymbol{b}_{k,h} = \boldsymbol{b}_{k-1,h} + \boldsymbol{\phi}(s'_{k,h+1}|s_{k,h}, a_{k,h})\boldsymbol{\delta}_{s_{k,h+1}}(s'_{k,h+1})$.
10:        $\boldsymbol{\theta}_{k,h} = \boldsymbol{M}^{-1}_{k,h}\boldsymbol{b}_{k,h}$.
11:     **end for**
12:     Compute upper occupancy bound: $u_k(s_{k,h}, a_{k,h}) = \text{COMP-UOB}(\pi_k, s_{k,h}, a_{k,h}, \mathcal{P}_k), \forall h$.
13:     Construct loss estimators for all $(s, a)$: $\widehat{\ell}_k(s, a) = \frac{\ell_k(s,a)}{u_k(s,a)+\gamma}\mathbb{I}_k\{s, a\}$.
14:     Update transition confidence set $\mathcal{P}_{k+1}$ based on Eq. (3).
15:     Compute occupancy measure: $\widehat{q}_{k+1} = \underset{q \in \Delta(\mathcal{P}_{k+1})}{\text{argmin}} \eta \left\langle q, \widehat{\ell}_k \right\rangle + D_F(q, \widehat{q}_k)$.
16:     Update policy $\pi_{k+1} = \pi^{\widehat{q}_{k+1}}$.
17: **end for**

---

## 4.1 CONFIDENCE SETS

One of the main difficulties in learning MDPs comes from the unknown transition $P$. To deal with this problem, a natural way is to construct its estimator together with the corresponding confidence set. Let $\boldsymbol{\phi}_V(s, a) = \sum_{s'} \boldsymbol{\phi}(s'|s, a)V(s')$. With the observation that $P_h(\cdot|s, a)^{\top}V_{k,h+1} = \sum_{s' \in \mathcal{S}} V_{k,h+1}(s')\langle\boldsymbol{\phi}(s'|s, a), \boldsymbol{\theta}^*_h\rangle = \langle\boldsymbol{\phi}_{V_{k,h+1}}(s, a), \boldsymbol{\theta}^*_h\rangle$, existing works studying linear mixture MDPs seek to learn $\boldsymbol{\theta}^*_h$ using $\boldsymbol{\phi}_{V_{k,h+1}}(s_{k,h}, a_{k,h})$ as feature and $V_{k,h+1}(s_{k,h+1})$ as the regression target (Ayoub et al., 2020; Cai et al., 2020). Particularly, they construct the estimator $\boldsymbol{\theta}_{k,h}$ of $\boldsymbol{\theta}^*_h$ as

$$\boldsymbol{\theta}_{k,h} = \underset{\boldsymbol{\theta} \in \mathbb{R}^d}{\text{argmin}} \sum_{i=1}^{k} \left[\left\langle \boldsymbol{\phi}_{V_{i,h+1}}(s_{i,h}, a_{i,h}), \boldsymbol{\theta}\right\rangle - V_{i,h+1}(s_{i,h+1})\right]^2 + \lambda\|\boldsymbol{\theta}\|^2_2.$$

Zhou et al. (2021); He et al. (2022) also use a similar method but further incorporate the estiamted variance information to gain a sharper confidence set. This method is termed as the *value-targeted regression* (VTR) (Ayoub et al., 2020; Cai et al., 2020; Zhou et al., 2021; He et al., 2022), which is critical to construct the optimistic estimator $Q_{k+1,h}(\cdot, \cdot)$ of the optimal action-value function $Q^*(\cdot, \cdot)$ and lead to the final regret guarantee.

However, though VTR is popular in previous works studying linear mixture MDPs (Ayoub et al., 2020; Cai et al., 2020; He et al., 2021a; Zhou et al., 2021; He et al., 2022; Wu et al., 2022; Chen et al., 2022b; Min et al., 2022), including the information of the state-value function $V_{i,h}(\cdot)$ in the regression makes this method hard to control the estimation error of the occupancy measure coming from the unknown transition $P$. To overcome this challenge, we seek a different way, in which $\boldsymbol{\theta}^*_h$ is learned by directly using the vanilla transition information.

Specifically, let $\boldsymbol{\Phi}_{s,a} \in \mathbb{R}^{d \times S}$ with $\boldsymbol{\Phi}_{s,a}(:, s') = \boldsymbol{\phi}(s'|s, a)$ and $\boldsymbol{\delta}_s \in \{0, 1\}^S$ be the Dirac measure at $s$ (*i.e.*, an one-hot vector with the one entry at $s$). To learn $\boldsymbol{\theta}^*_h$ from the transition information, one may consider using $\boldsymbol{\Phi}_{s_{k,h}, a_{k,h}}$ as feature and $\boldsymbol{\delta}_{s_{k,h+1}}$ as the regression target. Specifically, $\boldsymbol{\theta}_{k,h}$ could be taken as the solution of the following regularized linear regression problem:

$$\boldsymbol{\theta}_{k,h} = \underset{\boldsymbol{\theta} \in \mathbb{R}^d}{\text{argmin}} \sum_{i=1}^{k} \|\boldsymbol{\Phi}^{\top}_{s_{i,h}, a_{i,h}}\boldsymbol{\theta} - \boldsymbol{\delta}_{s_{i,h+1}}\|^2_2 + \lambda\|\boldsymbol{\theta}\|^2_2.$$

However, one obstacle still remains to be solved. Particularly, let $\boldsymbol{\eta}_{i,h} = P_h(\cdot|s_{i,h}, a_{i,h}) - \boldsymbol{\delta}_{s_{i,h+1}}$ be the noise at episode $i$ and stage $h$. Then it is clear that $\boldsymbol{\eta}_{i,h} \in [-1,1]^S$, $\mathbb{E}_{i,h}[\boldsymbol{\eta}_{i,h}] = \mathbf{0}$ and $\sum_{s \in \mathcal{S}} \boldsymbol{\eta}_{i,h}(s) = 0$. Therefore, conditioning on $\mathcal{F}_{i,h}$, the noise $\boldsymbol{\eta}_{i,h}(s)$ at each state $s$ is 1-subgaussian but they are not independent. In this way, one is still not able to establish an ellipsoid confidence set for $\boldsymbol{\theta}_{k,h}$ using the self-normalized concentration for vector-valued martingales (Abbasi-Yadkori et al., 2011). To further address this issue, we propose to use the transition information of only one state $s'_{i,h+1}$ in the next layer, which we call the *imaginary* next state. Note that the imaginary next state $s'_{i,h+1}$ is not necessary to be the *true* next state $s_{i,h+1}$ experienced by the learner. More specifically, we construct the estimator $\boldsymbol{\theta}_{k,h}$ of $\boldsymbol{\theta}_h^*$ via solving

$$\boldsymbol{\theta}_{k,h} = \underset{\boldsymbol{\theta} \in \mathbb{R}^d}{\operatorname{argmin}} \sum_{i=1}^k \left[ \left\langle \boldsymbol{\phi}\left(s'_{i,h+1}|s_{i,h}, a_{i,h}\right), \boldsymbol{\theta}\right\rangle - \boldsymbol{\delta}_{s_{i,h+1}}(s'_{i,h+1})\right]^2 + \lambda \|\boldsymbol{\theta}\|_2^2.$$

The closed-form solution of the above display is $\boldsymbol{\theta}_{k,h} = \boldsymbol{M}_{k,h}^{-1} \boldsymbol{b}_{k,h}$, where $\boldsymbol{M}_{k,h} = \sum_{i=1}^k \boldsymbol{\phi}(s'_{i,h+1}|s_{i,h}, a_{i,h})\boldsymbol{\phi}(s'_{i,h+1}|s_{i,h}, a_{i,h})^\top + \lambda \boldsymbol{I}$ is the feature covariance matrix at episode $k$ and stage $h$ and $\boldsymbol{b}_{k,h} = \sum_{i=1}^k \boldsymbol{\phi}(s'_{i,h+1}|s_{i,h}, a_{i,h})\boldsymbol{\delta}_{s_{i,h+1}}(s'_{i,h+1})$. The choice of $s'_{k,h+1}$ may be determined by the learner based on the information of previous steps up to observing $(s_{k,h}, a_{k,h})$. In particular, we choose $s'_{k,h+1}$ as

$$s'_{k,h+1} \in \operatorname{argmax}_{s \in \mathcal{S}_{h+1}} \|\boldsymbol{\phi}(s|s_{k,h}, a_{k,h})\|_{\boldsymbol{M}_{k-1,h}^{-1}}, \tag{2}$$

where the intuition is that the learner chooses to estimate the uncertainties of most uncertain states and hence controls the uncertainties of all the states in next layer. Based on the above construction of $\boldsymbol{\theta}_{k,h}$, we have its ellipsoid confidence set guaranteed by the following lemma.

**Lemma 1.** *Let $\delta \in (0, 1)$. Then for any $k \in \mathbb{N}$, and simultaneously for all $h = 0, \ldots, H-1$, with probability at least $1 - \delta$, it holds that $\boldsymbol{\theta}_h^* \in \mathcal{C}_{k,h}$, where $\mathcal{C}_{k,h} = \{\boldsymbol{\theta} \in \mathbb{R}^d : \|\boldsymbol{\theta} - \boldsymbol{\theta}_{k-1,h}\|_{\boldsymbol{M}_{k-1,h}} \leq \beta_{k,h}\}$ with $\beta_{k,h} = B\sqrt{\lambda} + \sqrt{2\ln(\frac{H}{\delta}) + \ln(\frac{\det(\boldsymbol{M}_{k-1,h})}{\lambda^d})}$.*

Note that the above lemma immediately implies that with probability $1 - \delta$, $P \in \mathcal{P}_k$, where $\mathcal{P}_k = \{\mathcal{P}_{k,h}\}_{h=0}^{H-1}$ and

$$\mathcal{P}_{k,h} = \{\widehat{P}_h : \exists \boldsymbol{\theta} \in \mathcal{C}_{k,h} \ s.t. \ \forall (s, a, s') \in \mathcal{S}_h \times \mathcal{A} \times \mathcal{S}_{h+1}, \widehat{P}_h(s'|s, a) = \boldsymbol{\theta}^\top \boldsymbol{\phi}(s'|s, a)\}. \tag{3}$$

### 4.2 LOSS ESTIMATORS AND ONLINE MIRROR DESCENT

**Loss Estimators** When learning the MDPs with known transition $P$, existing works consider constructing a conditionally unbiased estimator $\widehat{\ell}_k(s, a) = \frac{\ell_k(s,a)}{q_k(s,a)} \mathbb{I}_k\{s, a\}$ of the true loss function $\ell_k$ (Zimin & Neu, 2013; Jin & Luo, 2020), where $\mathbb{I}_k\{s, a\} = 1$ if $(s, a)$ is visited in episode $k$ and $\mathbb{I}_k\{s, a\} = 0$ otherwise. To further gain a high-probability bound, Ghasemi et al. (2021) extend the idea of *implicit exploration* in multi-armed bandits (Neu, 2015) and propose an optimistically biased loss estimator $\widehat{\ell}_k(s, a) = \frac{\ell_k(s,a)}{q_k(s,a) + \gamma} \mathbb{I}_k\{s, a\}$ with $\gamma > 0$ as the implicit exploration parameter. When transition $P$ is unknown, the true occupancy measure $q_k$ taken by the learner is also unknown, and the above loss estimators are no longer applicable. To tackle this problem, we use a loss estimator defined as $\widehat{\ell}_k(s, a) = \frac{\ell_k(s,a)}{u_k(s,a) + \gamma} \mathbb{I}_k\{s, a\}$ with $u_k(s, a) = \max_{\widehat{P} \in \mathcal{P}_k} q^{\widehat{P}, \pi_k}(s, a)$ termed as the *upper occupancy bound*, which is first proposed by Jin et al. (2020a). This loss estimator is also optimistically biased since $u_k(s, a) \geq q_k(s, a)$ given $P \in \mathcal{P}_k$ with high probability. Note that $u_k$ can be efficiently computed using COMP-UOB procedure of Jin et al. (2020a).

**Online Mirror Descent** To compute the updated occupancy measure in each episode, our algorithm follows the standard OMD framework. Since $\Delta(P)$ is unknown, following previous works (Rosenberg & Mansour, 2019b;a; Jin et al., 2020a), LSUOB-REPS runs OMD over the space of occupancy measures $\Delta(\mathcal{P}_{k+1})$ induced by the transition confidence set $\mathcal{P}_{k+1}$. Specifically, at the end of episode $k$, LSUOB-REPS updates the occupancy measure by solving

$$\widehat{q}_{k+1} = \underset{q \in \Delta(\mathcal{P}_{k+1})}{\operatorname{argmin}} \eta \left\langle q, \widehat{\ell}_k \right\rangle + D_F(q, \widehat{q}_k), \tag{4}$$

where $\widehat{\ell}_k$ is the biased loss estimator introduced above, $\eta > 0$ is the learning rate to be tuned later, $D_F(q, q') = \sum_{s,a} q(s,a) \ln \frac{q(s,a)}{q'(s,a)} - \sum_{s,a} (q(s,a) - q'(s,a))$ is the unnormalized KL-divergence, and the potential function $F(q) = \sum_{s,a} q(s,a) \ln q(s,a) - \sum_{s,a} q(s,a)$ is the unnormalized negative entropy. Besides, we note that Eq. (4) can be efficiently solved following the two-step procedure of OMD (Lattimore & Szepesvári, 2020). The concrete discussions are postponed to Appendix E. More comparisons between our method and previous methods are detailed in Appendix A.

## 5 ANALYSIS

In this section, we present the regret upper bound of our algorithm LSUOB-REPS, and a regret lower bound for learning adversarial linear mixture MDPs with unknown transition and bandit feedback.

### 5.1 REGRET UPPER BOUND

The regret upper bound of our algorithm LSUOB-REPS is guaranteed by the following theorem. Recall $d$ is the dimension of the feature mapping, $H$ is the episode length, $K$ is the number of episodes, $S$ and $A$ are the state and action space sizes, respectively.

**Theorem 1.** *For any adversarial linear mixture MDP $\mathcal{M} = (\mathcal{S}, \mathcal{A}, H, \{P_h\}_{h=0}^{H-1}, \{\ell_k\}_{k=1}^{K})$ satisfying Definition 1, by setting learning rate $\eta$ and implicit exploration parameter $\gamma$ as $\eta = \gamma = \sqrt{\frac{H \ln(HSA/\delta)}{KSA}}$, with probability at least $1 - 5\delta$, the regret of LSUOB-REPS is upper bounded by*

$$R(K) = O\left(dS^2\sqrt{K}\ln^2(K/\delta) + \sqrt{HSAK\ln(HSA/\delta)} + H\ln(H/\delta)\right).$$

*Proof sketch.* Let $q_k = q^{P,\pi_k}$. Following Jin et al. (2020a), we decompose the regret as

$$R(K) = \underbrace{\sum_{k=1}^{K}\left\langle \widehat{q}_k - q^*, \widehat{\ell}_k \right\rangle}_{\text{REG}} + \underbrace{\sum_{k=1}^{K}\left\langle q_k - \widehat{q}_k, \ell_k \right\rangle}_{\text{ERROR}} + \underbrace{\sum_{k=1}^{K}\left\langle \widehat{q}_k, \ell_k - \widehat{\ell}_k \right\rangle}_{\text{BIAS}_1} + \underbrace{\sum_{k=1}^{K}\left\langle q^*, \widehat{\ell}_k - \ell_k \right\rangle}_{\text{BIAS}_2}.$$

We bound each term in the above display as follows (see Appendix C.2 and Appendix C.3 for details). First, the REG term is the regret of the corresponding online optimization problem, which is directly controlled by the OMD and can be bounded by $O\left(\sqrt{HSAK\ln(HSA/\delta)} + H\ln(H/\delta)\right)$. Further, the BIAS$_2$ term measures the overestimation of the true losses by the constructed loss estimators, which can be bounded by $O\left(\sqrt{HSAK\ln(SA/\delta)}\right)$ via the concentration of the implicit exploration loss estimator (Lemma 1, Neu (2015); Lemma 11, Jin et al. (2020a)). Finally, the ERROR and BIAS$_1$ terms are closely related to the estimation error of the occupancy measure, which can be bounded by $O\left(S^2d\sqrt{K}\ln^2(K/\delta)\right)$ and $O\left(S^2d\sqrt{K}\ln^2(K/\delta) + \sqrt{HSAK\ln(HSA/\delta)}\right)$ respectively. Applying a union bound over the above bounds finishes the proof. □

**Remark 1.** *Ignoring logarithmic factors, LSUOB-REPS attains an $\widetilde{O}(dS^2\sqrt{K} + \sqrt{HSAK})$ regret guarantee when $K \geq H$. Compared with the regret bound $\widetilde{O}(dH^{3/2}\sqrt{K})$ of He et al. (2022) for the full-information feedback, our bound introduces the dependence on $S$ and $A$ and is worse than theirs since $S \geq H$ by the layered structure of MDPs. However, as we shall see in Section 5.2, incorporating the dependence on $S$ and $A$ into the regret bounds is inevitable at the cost of changing from the full-information feedback to the more challenging bandit feedback. Besides, when $dS \leq H\sqrt{A}$, the regret bound of LSUOB-REPS improves the regret bound $\widetilde{O}(HS\sqrt{AK})$ of Jin et al. (2020a).*

#### 5.1.1 BOUNDING THE OCCUPANCY MEASURE DIFFERENCE

To bound the ERROR and BIAS$_1$ terms, it is critical to control (a) the estimation error between $\widehat{q}_k$ and $q_k$; and (b) the estimation error between $u_k$ and $q_k$, both of which can be bounded by the following key technical lemma. We defer its proof to Appendix C.1.

**Lemma 2** (Occupancy measure difference for linear mixture MDPs). *For any collection of transition functions $\{P_k^s\}_{s \in \mathcal{S}}$ such that $P_k^s \in \mathcal{P}_k$ for all $s$, let $q_k^s = q^{P_k^s, \pi_k}$. If $\lambda \geq \delta$, with probability at least $1 - 2\delta$, it holds that $\sum_{k=1}^{K}\sum_{(s,a)\in\mathcal{S}\times\mathcal{A}} |q_k^s(s,a) - q_k(s,a)| = O\left(dS^2\sqrt{K}\ln^2(K/\delta)\right)$.*

**Remark 2.** *Comparing with the occupancy measure difference $\widetilde{O}\left(HS\sqrt{AK}\right)$ for tabular MDPs in Lemma 4 of Jin et al. (2020a), our bound $\widetilde{O}\left(dS^2\sqrt{K}\right)$ dose not have the dependence on $A$, though it is worse by a factor of $S$. The main hardness of simultaneously eliminating the dependence of the occupancy measure difference on $S$ and $A$ is that though the transition $P$ of a linear mixture MDP admits a linear structure, its occupancy measure still has a complicated recursive form: $q_k(s,a) = \pi_k(a|s)\left\langle \boldsymbol{\theta}^*_{h(s)-1}, \sum_{(s',a')\in\mathcal{S}_{h(s)-1}\times\mathcal{A}} q_k(s',a')\phi(s|s',a')\right\rangle$. We leave the investigation on whether it is possible to also eliminate the dependence on $S$ as our future work. Besides, we note that our bound for occupancy measure difference is not a straightforward extension of its tabular version of Jin et al. (2020a). Specifically, let $q_k^s(s,a|s_m)$ be the probability of visiting $(s,a)$ under the event that $s_m$ is visited in layer $m$. Jin et al. (2020a) decompose $q_k^s(s,a|s_m)$ as $(q_k^s(s,a|s_m) - q_k(s,a|s_m)) + q_k(s,a|s_m)$ and $(q_k^s(s,a|s_m) - q_k(s,a|s_m))$ will only contribute an $O(H^2S^2A\ln(KSA/\delta))$ term. However, in the linear function approximation setting, the above term will become a leading term with an $\widetilde{O}\left(H^2dS^2\sqrt{(d+S)K}\right)$ order. Hence we do not follow the decomposition of Jin et al. (2020a) and directly bound $q_k^s(s,a|s_m)$ instead.*

## 5.2 REGRET LOWER BOUND

In this subsection, we provide a regret lower bound for learning adversarial linear mixture MDPs with bandit feedback and unknown transition.

**Theorem 2.** *Suppose $A(H/2-1) \geq S-2-3H/4$, $(S-2-3H/4)A \geq 2(H/2-1)$, $S \geq 4+3H/2$, $2K \geq d$, $B \geq d/\sqrt{48K}$, and $H \geq 8$. Further assume $H/4$ and $\frac{S-2-3H/4}{H/2-1}$ are integers. Then for any algorithm, there exists an inhomogeneous, episodic $B$-bounded adversarial linear mixture MDP $\mathcal{M} = (\mathcal{S}, \mathcal{A}, H, \{P_h\}_{h=0}^{H-1}, \{\ell_k\}_{k=1}^K)$ satisfying Definition 1, such that the expected regret for this MDP is lower bounded by $\Omega(dH\sqrt{K} + \sqrt{HSAK})$.*

*Proof sketch.* At a high level, we construct an MDP instance such that it simultaneously makes the learner suffer regret by the unknown transition and the adversarial losses with bandit feedback. Specifically, we divide an episode into two phases, where the first and the second phase include the first $H/2+1$ layers and the last $H/2+1$ layers (layer $H/2$ belongs to both the first and the second phase). In the first phase, due to the unknown linear mixture transition functions, we can translate learning in this phase into simultaneously learning $H/4$ $d$-dimensional stochastic linear bandit problems with lower bound of order $\Omega(dH\sqrt{K})$. In the second phase, due to the adversarial losses with bandit feedback, we show that learning in this phase can be regarded as learning a combinatorial multi-armed bandit (CMAB) problem with semi-bandit feedback, the lower bound of which is $\Omega(\sqrt{HSAK})$. The proof is concluded by combining the bounds of the two phases. Please see Appendix D for the formal proof. $\qquad\square$

**Remark 3.** *The regret upper bound in Theorem 1 matches the lower bound in $d$, $K$, and $A$ up to logarithmic factors but looses a factor of $S^2/H$. The dependence of regret lower bound on $S$ and $A$ is inevitable since only the bandit feedback information of the adversarial losses is revealed to the learner and the loss function is nonstructural.*

## 6 CONCLUSIONS

In this work, we consider learning adversarial linear mixture MDPs with unknown transition and bandit feedback. We propose the first provably efficient algorithm LSUOB-REPS in this setting and prove that with high probability, its regret is upper bound by $\widetilde{O}(dS^2\sqrt{K} + \sqrt{HSAK})$, which only losses an extra $S^2/H$ factor compared with our proposed lower bound. To achieve this result, we propose a novel occupancy measure difference lemma for linear mixture MDPs by leveraging the transition information of the imaginary next state as the regression target, which might be of independent interest. One natural open problem is how to close the gap between the existing upper and lower bounds. Besides, our lower bound suggests that it is not possible to eliminate the dependence of the regret bound on $S$ and $A$ without any structural assumptions on the loss function. Generalizing the definition of linear mixture MDPs by further incorporating the structural assumption on the loss function (*e.g.*, the loss function is linear in the other unknown parameter) to eliminate the dependence on $S$ and $A$ also seems like an interesting future direction. We leave these extensions as future works.

ACKNOWLEDGMENTS

The corresponding author Shuai Li is supported by National Natural Science Foundation of China No. 62006151 and Shanghai Sailing Program. Baoxiang Wang is partially supported by National Natural Science Foundation of China (62106213, 72150002) and Shenzhen Science and Technology Program (RCBS20210609104356063, JCYJ20210324120011032).

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
