# OpenReview forum: "Learning Adversarial Linear Mixture Markov Decision Processes with Bandit Feedback and Unknown Transition"
_ICLR.cc/2023/Conference — ICLR 2023 poster_

### Official Review · Reviewer_ZddW · 2022-10-21

**Confidence:** 4
**Correctness:** 4
**Technical Novelty And Significance:** 3
**Empirical Novelty And Significance:** Not applicable
**Recommendation:** 6

**Clarity, Quality, Novelty And Reproducibility:**

Clarity: This paper is clear in general.Theorems have no ambiguity to me. Technical challenges and the corresponding solutions are described clearly. Related work is also discussed properly with now ambiguity.


Quality: The quality is good. This is a purely theoretical paper and the theorems are solid. The proof feels correct to me, though I did not check all the details.

Novelty: The paper is novel. The technical contributions, e.g. how to construct the confidence set of the transition parameter in the adversarial setting; how to choose the imaginary next state; are all good ideas in my opinion.


Reproducibility: This is purely theoretical work with no empirical result. The proof looks reproducible to me as it is written clearly and not very hard to read.


**Strength And Weaknesses:**

Strength:

1. This paper gives the first answer to a quite challenging problem, i.e., the adversarial linear MDPs. Both an upper bound and a lower bound (under certain assumptions) are given.

2. The paper is very well-written. This is a very technical paper. Techniques are presented with rather detailed explanations, making it easier for the readers to follow.

3. The paper has some novelty. The technical challenges are also discussed quite thoroughly, showing the technical contributions of the paper clear. For example, the imaginary next-state seems like a quite interesting idea to deal with the challenge of constructing a confidence ellipsoid when the classical self-normalized concentration technique cannot be directly applied.

---
Weakness:

There is no major weakness as far as I can tell. There are some comments though.
1. The dependence of the upper bound on $|S|$ is unpleasant under the linear mixture MDP setting.

2. The paper lacks a very detailed comparison with [1]. Although it seems that [1] studies tabular MDPs which is different from linear mixture MDP, the major idea of algorithmic design is very similar. Furthermore, in the case of representing a tabular MDP by a linear mixture MDP, we would have a relation $d = |S|^2 |A|$. In this case the bound in this paper would actually be worse than that of $1$ (but I can understand this since representing a tabular MDP by a linear mixture is inefficient. Still a more detailed comparison would be good to have).

3. There is no discussion on computational efficiency of the algorithm, so it might be inappropriate to call the algorithm ‘efficient’.

[1] Jin, Chi, et al. "Learning adversarial markov decision processes with bandit feedback and unknown transition." International Conference on Machine Learning. PMLR, 2020.

**Summary Of The Paper:**

This paper studies inhomogeneous adversarial linear mixture MDPs. It presents an efficient algorithm called LSUOB-REPS which provably achieves $\tilde{\mathcal{O}}(d S^2 \sqrt{K} + \sqrt{HSAK})$. The algorithm relies on occupancy measure based techniques, which maintains a confidence set of the unknown transition function and runs online mirror descent over the space of occupancy measures. One novelty is the construction of a specific linear regression objective function to solve for the transition parameter. To this end, this paper proposes to use an imaginary next state to construct the regression target. Furthermore, a lower bound of order $\Omega(dH\sqrt{K} + \sqrt{HSAK})$ is proven, which shows that the algorithm has close-to-optimal performance.

**Summary Of The Review:**

My current recommendation is accept.

The reasons are the following:

1. Solid contribution: this paper gives the answer to a quite challenging and important field of reinforcement learning theory.

2. Some novelty: to solve certain technical challenges, some novel ideas have been proposed.

---

> ### Author Response · Authors · 2022-11-15
> **Author Response 1**
>
> Thank you for your valuable comments and suggestions. We provide our response to each question in turn below.
>
> **Q1. "The dependence of the upper bound on $S$ is unpleasant under the linear mixture MDP setting."**
>
> In this paper, we follow the same linear mixture MDP definition in previous works, where only the transitions admit a linear structure and no structural assumptions are imposed on the loss functions. The regret bounds in previous works studying linear mixture MDPs do not have a dependence on $S$ (as well as $A$) since the loss functions are assumed to be deterministic and known [1,2,3] or adversarial with full-information feedback [4,5]. As shown in our lower bound, the dependence on $S$ and $A$ of regret bound for studying adversarial linear mixture MDPs in the more challenging bandit feedback setting is inevitable. Besides, as stated in Section 6, it is promising to eliminate the dependence on $S$ and $A$ by incorporating further structural assumptions on loss functions, and we leave this as our future work.
>
> **Q2. "The paper lacks a very detailed comparison ... Still a more detailed comparison would be good to have."**
>
> Regarding the algorithmic designs, both our algorithm and the previous occupancy measure-based algorithms [6,7,8] maintain a confidence set over the transitions and run OMD over the space of occupancy measures induced by all the plausible transitions within the confidence set. The key difference is that we devise an ellipsoid confidence set with the leverage of the linear structure of transitions, while their confidence sets are tailored to the tabular case. The other minor difference is that our occupancy measure is for state-action pairs while occupancy measures in [6,7,8] are for state-action-next-state triples, which leads to a slightly different optimization procedure of the implementation of OMD (see Appendix D for detailed discussions). To handle the adversarial losses with bandit feedback, we also adopt the idea of upper occupancy bound to construct an optimistically biased loss estimator, which is originally proposed in [6].
>
> In terms of the statistical results, compared with the occupancy measure difference for tabular MDPs in [6], our occupancy measure difference does not have a dependence on $A$, albeit with a slightly worse dependence on $S$. Hence, in an environment where $S$ is relatively smaller than $A$ (more specifically, $dS\leq H\sqrt{A}$), our regret bound improves the result in [6], due to the leverage of the inherent linear structure of transitions. When translating a tabular MDP into a linear mixture MDP by using one-hot feature mappings and setting $d=S^2A$, our regret bound becomes $\widetilde{O}(S^4A\sqrt{K})$ when $K\geq H$, and is worse than the regret bound in [6] by a factor of $S^3\sqrt{A}/H$. As the reviewer has recognized, the reason is that representing a tabular MDP as a linear mixture MDP simply using one-hot features is not efficient.
>
> We have now added the above comparisons regarding algorithmic designs in Section 4.2 in the revision of our paper (highlighted in green).

---

> ### Author Response · Authors · 2022-11-15
> **Author Response 2**
>
> **Q3. "There is no discussion on computational efficiency of the algorithm, so it might be inappropriate to call the algorithm ‘efficient’."**
>
> We would like to note that our algorithm can be implemented as efficiently as the algorithms in the tabular case [6,7,8]. In specific, the main computation issue only arises in the implementation of OMD (*i.e.*, solving Eq. (4)). Following the common two-step implementation procedure of OMD (see, *e.g.*, Chap 28 in [9]), Eq. (4) can be addressed by solving an unconstrained optimization problem in Eq. (5) which has a closed-form solution, and a projection problem in Eq. (6). The optimization procedure of the projection problem is similar with it in [6,7,8], where the minor difference coming from that our occupancy measure is for state-action pairs and theirs are for state-action-next-state triples, and hence the constraints over the occupancy measures are slightly different. More specifically, both the projection problems in our paper and in [6,7,8] are convex optimization problems with linear constraints and thus can be solved in polynomial time. Moreover, Slater’s condition in the primal problem holds, and therefore the strong duality holds. The strong duality implies that the projection problem can be solved more efficiently by solving its dual problem, which is also a convex problem with only non-negativity constraints (please see Appendix D for detailed discussions on the computation issue).
>
> We have added more discussions about the computation issue in Section 4.2 and Appendix D in the revision of our paper for clarity (highlighted in green).
>
>
>
> [1] Zhou, D., Gu, Q., & Szepesvari, C. (2021, July). Nearly minimax optimal reinforcement learning for linear mixture markov decision processes. In *Conference on Learning Theory* (pp. 4532-4576). PMLR.
>
> [2] Zhou, D., He, J., & Gu, Q. (2021, July). Provably efficient reinforcement learning for discounted mdps with feature mapping. In *International Conference on Machine Learning* (pp. 12793-12802). PMLR.
>
> [3] Ayoub, A., Jia, Z., Szepesvari, C., Wang, M., & Yang, L. (2020, November). Model-based reinforcement learning with value-targeted regression. In *International Conference on Machine Learning* (pp. 463-474). PMLR.
>
> [4] Cai, Q., Yang, Z., Jin, C., & Wang, Z. (2020, November). Provably efficient exploration in policy optimization. In *International Conference on Machine Learning* (pp. 1283-1294). PMLR.
>
> [5] He, J., Zhou, D., & Gu, Q. (2022, May). Near-optimal Policy Optimization Algorithms for Learning Adversarial Linear Mixture MDPs. In *International Conference on Artificial Intelligence and Statistics* (pp. 4259-4280). PMLR.
>
> [6] Jin, C., Jin, T., Luo, H., Sra, S., & Yu, T. (2020, November). Learning adversarial markov decision processes with bandit feedback and unknown transition. In *International Conference on Machine Learning* (pp. 4860-4869). PMLR.
>
> [7] Rosenberg, A., & Mansour, Y. (2019, May). Online convex optimization in adversarial markov decision processes. In *International Conference on Machine Learning* (pp. 5478-5486). PMLR.
>
> [8] Rosenberg, A., & Mansour, Y. (2019). Online stochastic shortest path with bandit feedback and unknown transition function. *Advances in Neural Information Processing Systems*, *32*.
>
> [9] Lattimore, T., & Szepesvári, C. (2020). *Bandit algorithms*. Cambridge University Press.

---

> ### Author Response · Authors · 2022-12-02
> **Author Response 3**
>
> Dear Reviewer ZddW:
>
> We thank you once again for your careful reading of our paper and your constructive comments and suggestions. We have given more discussions about our results, the computation issue, and more comparisons of the algorithm design with previous works. We have also revised our paper accordingly. As there are only about 10 days to the end of the second discussion stage, we will appreciate it very much if you could let us know whether all your questions are addressed. We are also more than happy to answer any further questions.

---

### Official Review · Reviewer_Buht · 2022-10-23

**Confidence:** 2
**Correctness:** 3
**Technical Novelty And Significance:** 3
**Empirical Novelty And Significance:** Not applicable
**Recommendation:** 6

**Clarity, Quality, Novelty And Reproducibility:**

Clarity: The paper is generally well-written and easy to follow. The literature review and comparison with existing works are sufficient.

Novelty: The analysis in the paper is sufficiently novel. The analysis utilizes a novel occupancy measure difference (Lemma 2), which is a non-trivial extension from Jin et al. (2020b). Although the tightness of Lemma 2 is unknown, it may be of independent interest and useful for the community.

Summary Of The Review: The paper studies adversarially linear mixture MDPs. The main results are interesting with a relatively novel analysis. However, the implications of the main results are unclear: what did we gain from the linear MDPs assumptions if the upper bound still depends on $S$?


**Strength And Weaknesses:**

Strengths:

(+) The upper bound is accompanied by a near-matching lower bound. The authors also justify the dependence on $S$ and $A$ via the lower bound.

(+) The use of a novel occupancy measure difference lemma (Appendix B.1).

Weaknesses:

(-) The bound depends on $S$, which is unusual and undesirable given the linear mixture MDPs assumption.

(-) The writing can be improved, especially the main results.

Detailed comments:

- I am familiar with statistical RL theory, but not so much in the adversarial setting. Hence, I will comment more on the writing/correctness aspects of the paper rather than the novelty/contribution.

- (main concern) Since the upper bound in Theorem 1 has a dependence on both $S$ and $A$, would this be just the tabular setting? I don’t quite understand where the linear approximation comes in when we still have the dependence on $S$.

- I encourage the authors to make it more explicit which elements are novel, particularly in your algorithm design. For example, how does your proposed algorithm differ from Jin et al. (2020b)?

- The main results (Theorem 1 and 2) should be made more detailed and self-contained. I encourage the authors to at least briefly introduce the settings and various relevant notations in the Theorems.

- The lower bound contains a term $dH\sqrt{K}$, which is linear in the horizon $H$. However, the upper bound only has a $\sqrt{H}$ dependence. Would this lead to a contradiction, or am I missing something here?

**Summary Of The Paper:**

The paper studies adversarial linear mixture MDPs with unknown transition and bandit feedback. The authors propose an algorithm with an upper bound $\tilde{O}(dS^2 \sqrt{K} + \sqrt{HSAK})$, and prove an almost near-matching lower bound (in $d, K, A$). The authors highlight the use of a novel occupancy measure difference lemma for linear mixture MDPs, which might be of independent interest.

**Summary Of The Review:**

The paper studies adversarially linear mixture MDPs. The main results are interesting with a relatively novel analysis. However, there is little innovation in the algorithm design, and the implications of the main results are unclear: what did we gain from the linear MDPs assumptions if the upper bound still depends on $S$?

---

> ### Author Response · Authors · 2022-11-15
> **Author Response 1**
>
> Thank you for your valuable comments and suggestions. We provide our response to each question below.
>
> **Q1. "The bound depends on $S$, which is unusual and undesirable given the linear mixture MDPs assumption"**
>
> In this paper, we follow the same linear mixture MDP definition in previous works, where only the transitions admit a linear structure and no structural assumptions are imposed on the loss functions. The regret bounds in previous works studying linear mixture MDPs do not have a dependence on $S$ (as well as $A$) since the loss functions are assumed to be deterministic and known [1,2,3] or adversarial with full-information feedback [4,5]. As shown in our lower bound, the dependence on $S$ and $A$ of regret bound for studying adversarial linear mixture MDPs in the more challenging bandit feedback setting is inevitable. Besides, as stated in Section 6, it is promising to eliminate the dependence on $S$ and $A$ by incorporating further structural assumptions on loss functions, and we leave this as our future work.
>
> **Q2. "Since the upper bound in Theorem 1 has a dependence on both $S$ and $A$, ... when we still have the dependence on $S$."**
>
> We would like to remark that our setting is *not* simply the tabular setting. Please see our response to Q1 for the reason why our bounds have a dependence on $S$ and $A$. Compared with the occupancy measure difference for tabular MDPs in [6], our occupancy measure difference does not have a dependence on $A$, albeit with a slightly worse dependence on $S$. Hence, in an environment where state space size $S$ is relatively smaller than the action space size $A$ (more specifically, $dS\leq H\sqrt{A}$), our regret upper bound is smaller than that in [6], due to the leverage of the inherent linear structure of transitions.
>
> **Q3. "I encourage the authors to make it more explicit which elements are novel, particularly in your algorithm design. "**
>
> Our algorithm is novel in that (a) we propose a new regression scheme with the leverage of the linear structure of the transitions; and (b) we devise the first ellipsoid confidence set in occupancy measure-based methods for RL with linear function approximation.
>
> Specifically, compared with previous works studying adversarial linear mixture MDPs [4,5], which use policy optimization-based methods and require full-information feedback, we propose the first occupancy measure-based method for learning adversarial linear mixture MDPs and our method can work under the bandit feedback setting. To construct the confidence set of transitions and hence occupancy measures, we propose to use the vanilla transition information of certain states in the next layer, which we call the imaginary next-states. Our regression scheme significantly departs from the value-targeted regression (VTR) scheme in [4,5] because we use different regression targets and different features.
>
> Compared with the works studying adversarial tabular RL using occupancy measure-based methods [6,7,8], both their algorithms and our algorithm share the similar idea of running OMD over the set of all the statistically plausible occupancy measures induced by the transitions within the transition confidence set. The main difference between our algorithm and previous ones is that we maintain an ellipsoid confidence set by leveraging the linear structure of the transitions, and previous methods are all tailored to the tabular case. Our algorithm is the first to maintain a confidence set over the occupancy measures for RL with linear function approximation, which is non-trivial (see the discussions in Section 5 in [9]).
>
> We have now added more discussions about the difference between the algorithm design in our paper and in previous works in Section 4.2 for clarity (highlighted in green).
>
> **Q4. "The main results (Theorem 1 and 2) should be made more detailed and self-contained."**
>
> We have now recalled the basic notations before Theorem 1, introduced the parameters used by our algorithm in Theorem 1, and briefly recalled our setting in both Theorem 1 and Theorem 2 for better readability (highlighted in green).

---

> ### Author Response · Authors · 2022-11-15
> **Author Response 2**
>
> **Q5. "The lower bound contains a term $dH\sqrt{K}$, ... Would this lead to a contradiction, or am I missing something here?"**
>
> Note that we adopt the layered structure assumption of MDP (see Section 3 for details), and each layer has at least one state. This implies that $S\geq H+1$ and there are no contradictions between our upper bound $\widetilde{O}\left(d S^2 \sqrt{K}+\sqrt{H S A K}\right)$ and lower bound $\Omega(d H \sqrt{K}+\sqrt{H S A K})$.
>
> **Q6. "there is little innovation in the algorithm design"**
>
> Please see our response to Q3 for the innovations of our algorithm.
>
> **Q7. "what did we gain from the linear MDPs assumptions if the upper bound still depends on $S$?"**
>
> The implication is that we now have a better understanding of adversarial RL with both linear function approximation and adversarial losses. Specifically, our results show that with the leverage of the linear structure over the transitions, we can further improve the results obtained using algorithms tailored to the tabular case when $S$ is relatively smaller than $A$ (see our response to Q2 for details). The second conclusion is that it is necessary to generalize the current definition of linear mixture MDPs by further incorporating structural assumptions over the loss functions to eliminate the dependence of regret bounds on both $S$ and $A$ in adversarial linear mixture MDPs. We leave it as our future work.
>
>
>
> [1] Zhou, D., Gu, Q., \& Szepesvari, C. (2021, July). Nearly minimax optimal reinforcement learning for linear mixture markov decision processes. In *Conference on Learning Theory* (pp. 4532-4576). PMLR.
>
> [2] Zhou, D., He, J., \& Gu, Q. (2021, July). Provably efficient reinforcement learning for discounted mdps with feature mapping. In *International Conference on Machine Learning* (pp. 12793-12802). PMLR.
>
> [3] Ayoub, A., Jia, Z., Szepesvari, C., Wang, M., \& Yang, L. (2020, November). Model-based reinforcement learning with value-targeted regression. In *International Conference on Machine Learning* (pp. 463-474). PMLR.
>
> [4] Cai, Q., Yang, Z., Jin, C., \& Wang, Z. (2020, November). Provably efficient exploration in policy optimization. In *International Conference on Machine Learning* (pp. 1283-1294). PMLR.
>
> [5] He, J., Zhou, D., \& Gu, Q. (2022, May). Near-optimal Policy Optimization Algorithms for Learning Adversarial Linear Mixture MDPs. In *International Conference on Artificial Intelligence and Statistics* (pp. 4259-4280). PMLR.
>
> [6] Jin, C., Jin, T., Luo, H., Sra, S., \& Yu, T. (2020, November). Learning adversarial markov decision processes with bandit feedback and unknown transition. In *International Conference on Machine Learning* (pp. 4860-4869). PMLR.
>
> [7] Rosenberg, A., \& Mansour, Y. (2019, May). Online convex optimization in adversarial markov decision processes. In *International Conference on Machine Learning* (pp. 5478-5486). PMLR.
>
> [8] Rosenberg, A., \& Mansour, Y. (2019). Online stochastic shortest path with bandit feedback and unknown transition function. *Advances in Neural Information Processing Systems*, *32*.
>
> [9] Neu, G., \& Olkhovskaya, J. (2021). Online learning in MDPs with linear function approximation and bandit feedback. *Advances in Neural Information Processing Systems*, *34*, 10407-10417.

---

> ### Comment · Reviewer_Buht · 2022-11-24
> **Post Rebuttal**
>
> I would like to thank the authors for their detailed responses. After reading the rebuttal, I believe the paper is interesting and the technical analysis is sufficiently novel.
>
> However, I am still a bit concerned about the significance of the paper. The benefit of the linear MDPs assumption and the proposed algorithm only comes in when we have $|S| \leq |A|$, which usually does not hold in practice.
>
> That being said, I agree with the authors that the paper provides a better understanding of adversarial RL with both linear function approximation and adversarial losses from a theoretical point of view. This can be of interest to the statistical RL community.
>
> For the reasons above, I will keep my original score and would like to recommend acceptance of this paper.
>
> Warmest regards,
> Reviewer.

---

> > ### Author Response · Authors · 2022-11-25
> > **Author Response 3**
> >
> > We thank you once again for your insightful and valuable review of our paper!

---

### Official Review · Reviewer_knov · 2022-10-24

**Confidence:** 3
**Correctness:** 4
**Technical Novelty And Significance:** 3
**Empirical Novelty And Significance:** Not applicable
**Recommendation:** 6

**Clarity, Quality, Novelty And Reproducibility:**

The paper is overall well-written and easy to read. But as I mentioned above, I feel there are some technical novelties that are not clearly explained in the main paper. Specifically,

1\ When talking about how to choose the next state in Eqn.(2), the author also mentioned: " for some technical reasons". But from my perspective it is not that trivial, my understanding is that it is like some "implicit exploration", choosing to maximize the most uncertain states? I would like to see a little more discussion on that.

2\ In remark 2, you said that "it is not a straightforward extension, even given the transition estimator obtained using the imaginary next-states." So my understanding is that, besides the imaginary next-states techniques, there are some other techniques that help you to further shave the dependency. I would like to see some discussion in the main paper.

3\ I personally think section 4.2 can be more concise since it basically gives a summary of the existing standard techniques.




**Strength And Weaknesses:**

Strength: The results are significant. The techniques seem novel to me -- (1) the use of " imaginary next state" is novel (2) the further improvement on H, S compared to Jin et al. (2020b) seems also interesting (but it is not very clearly stated)

Weakness: Some of the technique contributions are not clearly stated in the paper.

**Summary Of The Paper:**

This paper considers the adversarial linear adversarial MDP and proposes the first efficient algorithm that achieves $\sqrt{K}$ bound. Moreover, they also give a lower bound. Their techniques are based on the occupancy measurements which is initially proposed by Jin et al. (2020b). Compared to previous techniques, this paper adopts some nontrivial extensions, the main one, as stated in the paper, is the use of a carefully chosen " imaginary next state", which solves the dependence between the actual next states and thus allows the usage of regular linear regression to estimate the $\theta$

**Summary Of The Review:**

Overall I think it has good results and relatively novel techniques. But the explanation of some techniques seems not very clear to me. So I give 6 and would like to raise my score if I can better understand the novelty of the paper.

---

> ### Author Response · Authors · 2022-11-15
> **Author Response 1**
>
> Thank you for your valuable comments and suggestions. We provide our response to each question below.
>
> **Q1. "When talking about how to choose the next state ... I would like to see a little more discussion on that."**
>
> Yes, it is indeed a kind of *exploration* of the states in the sense that the learner chooses to estimate the uncertainties of the most uncertain states. In this way, we are able to bound the uncertainties of all the states in the next layer via the elliptical potential lemma, which is critical to control the occupancy measure difference in Lemma 2. We have added more explanations about the intuitions of the choice of imaginary next-states in Section 4.1 in the revision of our paper (highlighted in green).
>
> **Q2. "besides the imaginary next-states techniques, ... I would like to see some discussion in the main paper."**
>
> Both our method and the methods in [1] share the same basic idea to bound the occupancy measure difference, in which the occupancy measure difference is eventually transformed into the difference between the estimated transition and the true transition and hence controlled by the transition confidence set. However, since we use an ellipsoid confidence set in linear mixture MDPs while their confidence set is tailed to the tabular case, improvements are needed to achieve our result $\widetilde{O}\left(d S^2 \sqrt{K} \right)$ for occupancy measure difference.
>
> In specific, let $q_k^s(\cdot,\cdot)$ be the occupancy measure induced by some transition function $P^s_k$ associated with $s$ and let $q_k^s(\cdot):=\sum_{a\in\mathcal{A}}q_k^s(\cdot,a)$. Also, denote by $q_k^s\left(s,a | s_m\right)$  the probability of visiting $(s,a)$ under the event that $s_m$ is visited in layer $m$. Since
> $\sum_{(s,a)\in \mathcal{S}_h\times\mathcal{A}}q_k^s\left(s,a | s_m\right)$  is involved  in the proof of the occupancy  measure difference, bounding this term will lead to a $S_h$ factor
>
> (note that $\set{q^s_k(s)}_{s\in\mathcal{S}_h}$ is not necessarily a probability measure over  $\mathcal{S}_h$
>
> because $\set{P_k^s}_{s \in \mathcal{S}_h}$ may vary).
>
> In tabular case, to avoid this $S_h$ factor, we can decompose $q_k^s\left(s,a | s_m\right)$ into $(q_k^s\left(s,a | s_m\right)-q_k\left(s,a | s_m\right))+q_k\left(s,a | s_m\right)$ as [1]. Then combined with the tabular transition confidence set (*e.g.*, the Bernstein-style confidence set used in [1]),
> $
> \sum_{k,s,a}\sum_{1\leq m<h(s)}\sum_{s_m\in\mathcal{S}_m}(q_k^s\left(s,a | s_m\right)-q_k\left(s,a | s_m\right))
> $
>
> will only have $O(\log K)$ dependence on the episode number $K$, albeit with a worse multiplicative factor of order $O(H^2S^2A)$. However, in the linear function approximation setting and using the transition estimator based on imaginary next-states, directly following the decomposition in [1], $\sum_{k,s,a}\sum_{1\leq m<h(s)}\sum_{s_m\in\mathcal{S}_m}(q_k^s\left(s,a | s_m\right)-q_k\left(s,a | s_m\right))$  will become a leading term and have an order of $\widetilde{O}\left(H^2 dS^2  \sqrt{(d+S) K}\right)$. Therefore, we do not adopt this decomposition and directly bound $q_k^s\left(s,a | s_m\right)$ together with other terms instead, which finally leads to our $\widetilde{O}\left(d S^2 \sqrt{K} \right)$ bound.
>
> We have now added more discussions about this point in Section 5.1.1 in the revision of our paper (highlighted in green).
>
> **Q3. "I personally think section 4.2 can be more concise ..."**
>
> We have revised Section 4.2 accordingly to make it more concise (highlighted in green).
>
> [1] Jin, C., Jin, T., Luo, H., Sra, S., & Yu, T. (2020, November). Learning adversarial markov decision processes with bandit feedback and unknown transition. In *International Conference on Machine Learning* (pp. 4860-4869). PMLR.

---

> > ### Comment · Reviewer_knov · 2022-12-05
> > **Feedback**
> >
> > I think the explanation are good to me and I will keep my score 6.

---

> ### Author Response · Authors · 2022-12-02
> **Author Response 2**
>
> Dear Reviewer knov:
>
> We thank you once again for your careful reading of our paper and your constructive comments and suggestions. We have given more discussions about the design of our algorithm as well as the theoretical results and revised our paper accordingly. As there are only about 10 days to the end of the second discussion stage, we will appreciate it very much if you could let us know whether all your questions are addressed. We are also more than happy to answer any further questions.

---

### Official Review · Reviewer_1UrW · 2022-11-02

**Confidence:** 3
**Correctness:** 3
**Technical Novelty And Significance:** 2
**Empirical Novelty And Significance:** Not applicable
**Recommendation:** 5

**Clarity, Quality, Novelty And Reproducibility:**


The writing is clear and of good quality. No apparent typos are noticed.

**Strength And Weaknesses:**


### Strength

1. This paper studies a reasonable setting of "adversarial" RL with linear function approximation, and designs and proves the first algorithm with sqrt(K) regret.
2. This paper goes beyond the classical VTR (value-targeted regression) framework of analysis for linear mixture MDP, coming up with a novel technique (imaginary roll out of state) to cope the challenge arouse in adversarial setting.

### Weakness

1. There are many settings of linear RL theory besides linear mixture MDP, such as linear MDP. Also, recently, there are unified framework such as Bellman-rank and bilinear classes for RL with general function approximation. In particular, these frameworks include linear mixture MDP as their special cases. While the theory of RL with function approximation is certainly an important topic, it could become much less important when the scope is restricted to linear mixture MDP.
2. This paper does not prove tight lower bound to complete the theoretical research in their setting.


**Summary Of The Paper:**



The paper studies RL in linear mixture MDP with unknown transition, adversarial loss, and bandit feedback. This paper gives the first algorithm in this setting whose regret scales with sqrt(#episode) and supplements a lower bound, which matches the upper bound for the $d$ and $K$ dependency, where $K$ is #epsiode and $d$ is the ambient dimension of the linear mixture MDP.

**Summary Of The Review:**


This paper contains some novel results and techniques, but it could be made better if the authors could come up with more complete results.

---

> ### Author Response · Authors · 2022-11-15
> **Author Response 1**
>
> Thank you for your valuable comments and suggestions. We provide our response to each question below.
>
> **Q1. "There are many settings of linear RL theory besides linear mixture MDP, such as linear MDP.... it could become much less important when the scope is restricted to linear mixture MDP."**
>
> We think the topic of studying linear mixture MDPs is important, and it has gained lots of research attention in the community [1-5], though the setting of linear mixture MDPs is subsumed in RL with general function approximation such as bilinear classes. Besides, studying RL with general function approximation and adversarial losses is important but also challenging. Following previous works [4,5], we also take learning adversarial linear mixture MDPs as the first step. Moreover, the state-of-the-art algorithm learning adversarial linear MDPs with both unknown transition and bandit feedback only attains an $\widetilde{O}(K^{14/15})$ regret guarantee [6], and the results of linear MDPs and linear mixture MDPs do not imply each other due to different assumptions imposed on both transitions and loss functions.
>
> **Q2. "This paper does not prove tight lower bound to complete the theoretical research in their setting."**
>
> While our lower bound does not match our upper bound in this paper, we think our results are also significant and meaningful to the literature since we provide the first $\widetilde{O}(K)$ regret upper bound for RL with linear function approximation under unknown transition, adversarial losses and bandit feedback. Further, ignoring logarithmic factors, our upper bound matches the lower bound in $K$, $d$, $A$ and only losses an $S^2/H$ factor. Besides, we would like to note that studying adversarial RL with bandit feedback is challenging, and the gap between upper and lower bounds is still open even in tabular RL [7].
>
> [1] Zhou, D., Gu, Q., & Szepesvari, C. (2021, July). Nearly minimax optimal reinforcement learning for linear mixture markov decision processes. In *Conference on Learning Theory* (pp. 4532-4576). PMLR.
>
> [2] Zhou, D., He, J., & Gu, Q. (2021, July). Provably efficient reinforcement learning for discounted mdps with feature mapping. In *International Conference on Machine Learning* (pp. 12793-12802). PMLR.
>
> [3] Ayoub, A., Jia, Z., Szepesvari, C., Wang, M., & Yang, L. (2020, November). Model-based reinforcement learning with value-targeted regression. In *International Conference on Machine Learning* (pp. 463-474). PMLR.
>
> [4] Cai, Q., Yang, Z., Jin, C., & Wang, Z. (2020, November). Provably efficient exploration in policy optimization. In *International Conference on Machine Learning* (pp. 1283-1294). PMLR.
>
> [5] He, J., Zhou, D., & Gu, Q. (2022, May). Near-optimal Policy Optimization Algorithms for Learning Adversarial Linear Mixture MDPs. In *International Conference on Artificial Intelligence and Statistics* (pp. 4259-4280). PMLR.
>
> [6] Luo, H., Wei, C. Y., & Lee, C. W. (2021). Policy Optimization in Adversarial MDPs: Improved Exploration via Dilated Bonuses. *arXiv preprint arXiv:2107.08346*.
>
> [7] Jin, C., Jin, T., Luo, H., Sra, S., & Yu, T. (2020, November). Learning adversarial markov decision processes with bandit feedback and unknown transition. In *International Conference on Machine Learning* (pp. 4860-4869). PMLR.

---

> ### Author Response · Authors · 2022-12-02
> **Author Response 2**
>
> Dear Reviewer 1UrW:
>
> We thank you once again for your careful reading of our paper and your constructive comments and suggestions. As there are only about 10 days to the end of the second discussion stage, we will appreciate it very much if you could let us know whether all your concerns are addressed. We are also more than happy to answer any further questions.

---

> ### Author Response · Authors · 2022-12-09
> **Author Response 3**
>
> Dear Reviewer 1UrW,
>
> We thank you again for your tremendously valuable review of our paper. As the second discussion stage is coming to an end, please let us know if you have any questions about our responses or any further concerns. If not, we will appreciate it very much if you can consider improving your evaluation of our paper.
>
> Best, Authors.

---

### Decision · Program_Chairs · 2023-01-20

**Decision:**

Accept: poster

**Justification For Why Not Higher Score:**

Borderline, but reasonably good.

**Justification For Why Not Lower Score:**

Paper makes positive contribution.

**Metareview: Summary, Strengths And Weaknesses:**

The paper studies adversarial linear mixture MDPs with unknown transition and bandit feedback. The authors propose an algorithm with an upper bound of \tilde{O}(dS^s \sqrt{K} + \sqrt{HSAK}) where S is the size of the statespace, A the size of the action space, H the episode length and K the number of epsisodes. The algorithm is efficient (though a slightly restricted setting is considered by necessity) and there is a lower bound that matches in some factors. Some techniques in the paper might be of independent interest and although there are a lot of open problems remaining, the papers makes sufficient contributions to merit acceptance.

**Note From Pc:**

if the above contains the word "oral" or "spotlight" please see: "oral" presentation means -> notable-top-5% and "spotlight" means -> notable-top-25%. As stated in our emails, we are disassociating presentation type from AC recommendations